# Challenges and Opportunities of the Mediterranean Indigenous Bovine Populations: Analysis of the Different Production Systems in Algeria, Greece, and Tunisia

Aziza Mohamed-Brahmi [1,*,†] , Dimitrios Tsiokos [2,†] , Samia Ben Saïd [1] , Sofiane Boudalia [3] , Samir Smeti [4] , Aissam Bousbia [3] , Yassine Gueroui [5] , Ali Boudebbouz [3] , Maria Anastasiadou [2] and George K. Symeon [2,*]

1 Laboratoire de Durabilité des Systèmes de Production dans la Région du Nord-Ouest, Ecole Supérieure d'Agriculture du Kef, Université de Jendouba, Boulifa, Le Kef 7100, Tunisia; sabensaid@gmail.com
2 Research Institute of Animal Science, HAO-Demeter, 58100 Giannitsa, Greece; tsiokosd@gmail.com (D.T.); marmogeo@gmail.com (M.A.)
3 Laboratoire de Biologie, Eau et Environnement, Département d'Écologie et Génie de l'Environnement, Université 8 Mai 1945, Guelma BP 4010, Guelma 24000, Algeria; boudalia.sofiane@univ-guelma.dz (S.B.); bousbia.aissam@univ-guelma.dz (A.B.); boudebbouz.ali@univ-guelma.dz (A.B.)
4 Laboratoire de Productions Animales et Fourragères, INRA-Tunisia, Ariana 2049, Tunisia; sam_fsb@live.fr
5 Département des Sciences de la Nature et de la Vie, Université 8 Mai 1945, Guelma 24000, Algeria; gueroui.yassine@univ-guelma.dz
* Correspondence: mohamedaziza2003@yahoo.fr (A.M.-B.); gsymewn@yahoo.gr (G.K.S.); Tel.: +216-54-471316 (A.M.-B.); +30-2-382-031-700 (G.K.S.)
† These authors contributed equally to this work.

**Abstract:** The indigenous cattle populations are threatened by extinction in many countries of the Mediterranean area. The objective of this study is the analysis of local cattle breeds' production systems in Algeria, Greece, and Tunisia and the identification of their future challenges and opportunities. A total of 385 surveys were conducted in these study areas: central and northern Greece (43); northern and northwestern Tunisia (167), and northeastern Algeria (175). Data collected concerned socio-economic parameters as well as the production system's functionality, constraints, and opportunities. Results revealed an average farmers' age of 52.6 years old. The illiteracy rate is high, especially in Algeria (39%) and Tunisia (44%), where the farm size is relatively small with an average of 14 and four animals per farm, respectively. In Greece, much higher numbers were recorded (89 animals/farm). The average cultivated feedstuffs' area is larger in Greece (12.07 ha) and smaller in Algeria and Tunisia (6.11 and 2.88 ha, respectively). Feeding resources are based on rangelands. Farming systems are traditional extensive and complemented when needed. Milk and meat marketing vary throughout countries and are not well valorized. The main constraints are high feeding costs, low milk and meat prices, and absence of labeling. Local and local-crossbred bovine populations could be valorized based on their good adaptation criteria when applying convenient genetic and development strategies.

**Keywords:** indigenous; cattle; Mediterranean; opportunities; challenges

## 1. Introduction

The indigenous cattle production systems contribute to milk and meat supply and represent an essential source for many communities in rural areas in the Mediterranean countries [1]. These populations are extremely valuable both at the local and regional level since they combine unique qualities: a valuable locally adapted genetic pool, substantial income to the local economies, and added-value animal products. Nevertheless, their numbers are declining due to the preference of farmers toward foreign, more productive breeds. Especially in the Mediterranean countries, the indigenous cattle breeds' populations face continuous challenges such as fear of extinction, anarchic breeding schemes, and

harsh rearing conditions [2]. Generally, the global livestock sector is characterized by a growing contrast between livestock kept extensively by a large number of smallholders and pastoralists (600 million) in support of rural food security and livelihoods, and those kept in intensive commercial production systems [2,3]. In Southern Africa, over 90% of animal keepers are classified as smallholders and 75% of the farm animals, which largely consists of indigenous breeds, belong to the smallholder sector [4].

In Greece, the indigenous cattle populations have decreased to small numbers and are currently at risk of extinction, or already extinct, due to socio-economic reasons, geographic isolation, and crossbreeding with commercial breeds [5]; in 2020, four indigenous breeds were referred according to the Domestic Animal Diversity Information System of FAO (as provided by the Greek Ministry of Agriculture). They are used exclusively for meat production: the "Greek Red" (42,057 females), the "Vrahykeratiki" (9546 females), the "Katerinis" (728 females), and "Sykias" breed (2851 females) [6]. These animals are reared essentially in the mountainous grasslands. They are raised all year in the fields and housed only in extreme weather conditions. They are held in rough housing, and their dietary needs are covered mostly by grazing, while complementary feed is provided only in the winter. In past decades, the importation and use of foreign breeds and the disorganized breeding schemes have resulted in a great variety of phenotypes [1]. In the last 20 years, conservation programs have been set up to safeguard the indigenous populations.

In Tunisia, the indigenous cattle population with Iberian origin counts 191,920 females which are mainly (87%) localized in the north, especially in the mountainous area (120,000 heads). In this zone, indigenous cattle breeds contribute to 15–26% of the milk and meat production. This population has suffered from anarchic crossing, which affected its genetic structure [7]. Nevertheless, studies concerning this breed were mostly interested in genetic aspects. In fact, two breeds were identified: Atlas Brown and Blonde of Cap Bon. Moreover, the population of Atlas Brown has been declining over the years, and the population of the Blonde of Cap Bon is very limited, indicating that it is exposed to extinction [7]. The study of phenotypic variability based on a qualitative description of the characters showed that the differences between individuals are mainly manifested through the color and the general conformation of the animals.

Algerian indigenous cattle populations resemble the Atlas Brown, with pure bred animals still preserved in the mountainous regions. They are subdivided into several sub-populations, namely "Guelmoise", "Cheurfa", "Krouminiène", "Chelifienne", "Sétifienne", and "Djerba", which are clearly differentiated phenotypically [8]. These populations are characterized by their rusticity, and they constitute a very important socio-economic element, contributing to a large part to the feeding of the rural population [1,9]. Indeed, these populations have brought together qualities of adaptation to the harsh arid and semi-arid environment and to the food resources restriction [10–12]. Despite the perfect harmony between these indigenous cattle populations and their natural environment, productivity remains modest (1175 litter/cow/year [13]) both because of the often unfavorable rearing conditions and the low performance of the concerned breeds. Several trials for dairy intensification, based mainly on the importation of exotic breeds and the anarchic crossbreeding with the indigenous populations, led to a deterioration of the genetic structure of the dairy herd in Algeria, which resulted in a drastic fall in the numbers of local cattle. Thus, the percentage of indigenous cattle breeds' population is reduced from 82% of the total in 1986 to about 48% of the total in 2016 [14,15].

Therefore, the concept of the scientific cooperation project BOVISOL (Breeding and management practices of indigenous bovine breeds: Solutions toward a sustainable future) arose as a necessity between the partners (Algeria, Greece, and Tunisia) to preserve these populations by trying to find the best tools that will improve the production systems in terms of productivity and sustainability. The general aim of this project was to contribute to the sustainability of the indigenous bovine breeds' production systems by taking into account the adaptability of the animals to the local environment, the quality of the animal products, and the economic and cultural value of the systems. After all, as also very

well documented by [16], it is not an easy task to balance between the values of biodiversity, cultural heritage, and productivity, as these values are differently perceived by the stakeholders in the sector.

There were three specific objectives of this work:

- socio-economic identification of the breeders of the indigenous cattle populations in the different study areas.
- analytical description of the existing farm and breeding practices.
- identification of constraints and proposition of solutions that will promote the sustainability of these production systems in the context of the climate change challenges.

## 2. Materials and Methods

### 2.1. Location of the Study Areas, Farmers Sampling, and Data Collection

In each country, the local Data Protection Board (DPB) and the local Ethics Committee have approved experimental protocols. The study involved data collection from different farms, and participants were informed of the purpose of the project; they have given their consent for their participation (complete the survey questionnaire and/or provide a sample of the milk) and the use of data collected and generated for scientific publications.

Study areas concerned regions in the three countries where indigenous cattle breeding and rearing is usually practiced. The collected data comprised a total of 385 questionnaires answered by owners or people who are responsible on the farms for the indigenous or crossbred cattle randomly selected from different villages in the study area. In Greece, the study was carried out in the central and northern Greece regions of Thessaly, Macedonia, and Thrace from March 2018 to May 2019. These study areas are known for their high density of cattle population. Cattle farms were selected taking into account the representation of all major indigenous cattle breeds, farm sizes, and typical Greek geographical conditions. In Tunisia, the study was carried out from January 2019 to June 2021. Surveys were conducted in two regions located in the north and northwestern Tunisia: Sejnan (Bizerte) and Tabarka-Ain Draham (Jendouba). These regions are plains and mountainous areas known for the predominance of indigenous and crossbred indigenous cattle breeding. In Algeria, surveys were conducted in the region of Guelma, Skikda, Annaba, and Bordj Bou Arréridj in northeast of the country from June 2018 to May 2019. The region is characterized by a subhumid climate in the center and in the north and semi-arid in the south. The climate is mild and rainy in winter and hot and dry in summer.

In all three countries, data were collected through direct interviews, using a quasi-structured questionnaire and personal observation at each visit during the study period. The interviewers followed a participatory way, where breeders had been asked to provide demographic information regarding the age, the education level, economic activities, as well as data regarding the livestock management as well as breeding and feeding practices.

### 2.2. Statistical Analyses

Details on the farms' structure, the breeds, the animals' performances, production systems, and market channels were digitized in spreadsheets (MS EXCEL 2016) separately for each country and coded, entered, corrected, and validated by the research team in accordance to the common format of the three countries before being imported in IBM SPSS Statistics package version 25 (IBM SPSS, 2017). From the 91 initial variables produced from the questionnaires, 17 variables were removed either due to missing data from one or two of the countries (more than 50% missing values in one country) or containing information irrelevant to the present study, mainly because the farmers found it difficult to understand the meaning of the questions. New variables were computed, where necessary, by combining variables from the questionnaires in order to reduce the data presented and to produce clearer results. Analysis was proceeded with quality control, corrections, and further validations. Farms with missing values were removed from the database. The custom tables function was used in IBM SPSS Statistics package version 25 (IBM SPSS, 2017) in order to create tables presenting all the results between the countries, and the created tables were

imported in spreadsheets (MS EXCEL 2016) to produce the figures. Additionally, Pearson's chi-squared test was used to determine whether there was a statistically significant difference between the countries regarding the categorical variables and one-way analysis of variance (ANOVA) to determine whether there was a statistically significant difference between the countries regarding the continuous variables. A SWOT analysis was finally carried out relative to the sustainability of the indigenous cattle production systems in the study area, utilizing the data used in the previous parts of the work.

## 3. Results and Discussion

### 3.1. Farmers' Socio-Economic Identification

The main characteristics of farmers' identity are reported in Table 1. Results show that indigenous cattle farming is the main occupation for 82.3% of farmers, and 69% of them have a successor in the farm, which is a number that is greater in the North African countries (Table 1). Especially for Greece, the lower proportions of the existence of successors in the farms are a serious constraint factor, since particularly younger individuals reject traditional livestock farming because of the harsh working conditions and the low social status associated with this occupation [17]. The average age of the farmers is 52 years, and most of them are married (89%) with an average of three children.

**Table 1.** Socio-economic identification of farmers who participated in the survey.

| Country | Farmers' Average Age | Married Farmers | Number of Children | Successor in the Farm | Farmers' with More than 20 Years' Experience | Farmers Working Full Time in Livestock |
|---|---|---|---|---|---|---|
| | One-way ANOVA $F_{(2, 380)} = 4.582$, $p < 0.05$ | Chi-Square = 14.577; df (2); $p < 0.01$; Cramer's V = 0.195 | One-way ANOVA $F_{(2, 382)} = 32.615$, $p < 0.01$ | Chi-Square = 17.517; df (2); $p < 0.01$; Cramer's V = 0.213 | Chi-Square = 34.770; df (4); $p < 0.01$; Cramer's V = 0.215 | Chi-Square = 69.033; df (2); $p < 0.01$; Cramer's V = 0.423 |
| Algeria | 55.0 | 91.4% | 4.0 | 74.3% | 66.9% | 56.6% |
| Greece | 48.3 | 79.1% | 1.6 | 51.2% | 63.6% | 100.0% |
| Tunisia | 53.6 | 96.4% | 2.8 | 82.0% | 41.9% | 90.4% |
| Overall Average | 52.6 | 89.0% | 3.2 | 69.2% | 57.5% | 82.3% |

df: degree of freedom.

An attention-grabbing figure in the current study is the relatively high illiteracy rate, especially in Algeria (39%) and Tunisia (44%). Low literacy is a concept often observed in rural areas in Algeria and Tunisia, and it is partly explained by the farms' location in remote areas, without schools and cultural centers [18,19]. Even in Greece, where 80% of farmers have completed the basic education, only a very small percentage (5%) indicated that they have received some kind of training related to animal breeding. This is quite interesting, since it has been proven that there is a significant positive relationship between the level of farmers' education and the level of productivity [20], while value addition can also be promoted through training and capacity building [21].

In Tunisia and Algeria, more than 70% of the surveyed farmers opt for this profession due to heritage, while in Greece, the principal reason for this choice is the love for the profession (49%). Profit as a reason for practicing the profession varied throughout the three countries: 28% in Greece, 30% in Algeria, and only 5% in Tunisia, which was probably due to the limited productive performances of the indigenous cattle breeds (Chi-square = 275.687; df (18); $p < 0.01$; Cramer's V = 0.600) (Figure 1).

With respect to the reasons for choosing the indigenous versus the commercial breeds, the majority of farmers in Algeria and Tunisia (72.6% and 98.8%, respectively) chose the breeds' adaptation characteristics (Table 2). Moreover, more than 50% of the investigated farmers in the different countries indicated that they chose these breeds because of their productive performances, although there may be a misinterpretation between the performances (quantity of product produced) and robustness of the breeds. In Greece, where there are European and state funding conservation programs, 5% of the farmers mentioned

that they are practicing this activity due to the subsidies, which was a relatively low proportion that was expected to be higher. Even though the farmers do not admit it openly, it has been reported that the conservation of indigenous breeds may not be viable without economic support [22].

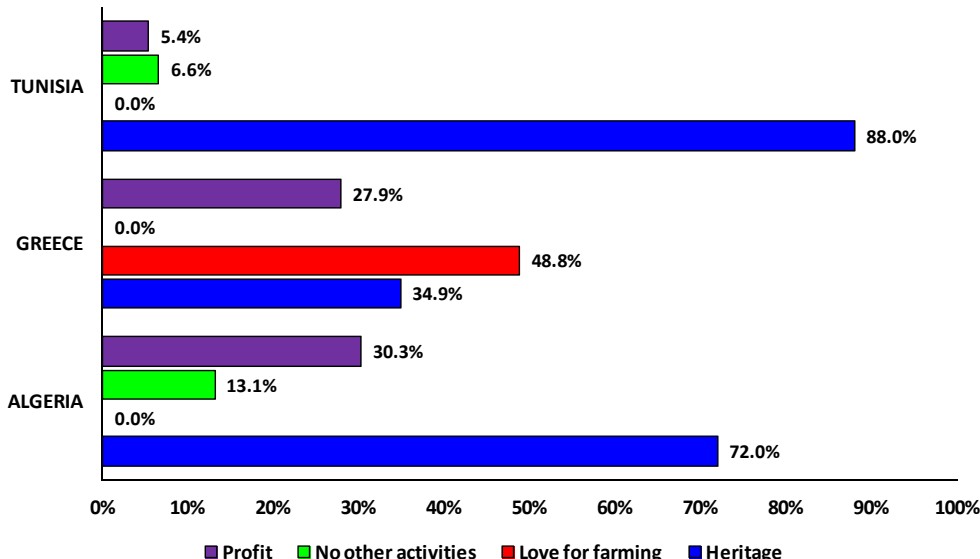

**Figure 1.** Reasons of practicing indigenous cattle farming in the study area.

**Table 2.** Reasons of choosing the indigenous cattle breeds (percentage of breeders).

|  | **Adaptation** | **Performances** | **Subsidies** |
|---|---|---|---|
| Algeria | 72.6% | 53.7% | 0.0% |
| Greece | 20.9% | 51.2% | 4.7% |
| Tunisia | 98.8% | 64.7% | 0.0% |

*3.2. General Characterization of the Indigenous Cattle Production Systems*

3.2.1. Farm Size and Type

Table 3 presents the main elements of farm sizing in the three studied countries. In terms of animal population, in Tunisia and Algeria, the farm size is relatively small with an average of four and 14 indigenous cows per farm, while in Greece, the respective figure is much higher (89 animals per farm). This is also depicted in the total farm area as well as the average number of cattle per hectare.

**Table 3.** Farm structure in the three countries.

|  |  | **Algeria** | **Greece** | **Tunisia** |
|---|---|---|---|---|
| Average cattle number per farm | One-way ANOVA F(2, 381) = 183.688, $p < 0.01$ | 13.97 (14.62) * | 88.90 (72.6) * | 3.76 (4.78) * |
| Total area of the farm (ha) | One-way ANOVA F(2, 336) = 19.801, $p < 0.01$ | 21.24 (16.17) * | 8.85 (13.29) * | 12.41 (11.87) * |
| Cattle per ha | One-way ANOVA F(2, 335) = 79.485, $p < 0.01$ | 0.72 (0.63) * | 15.53 (19.40) * | 0.91 (0.89) * |
| Area cultivated for feedstuffs (ha) | One-way ANOVA F(2, 351) = 14.883, $p < 0.01$ | 6.11 (5.82) * | 12.07 (12.55) * | 2.88 (3.26) * |
| Own cultivated area (%) | One-way ANOVA F(2,312) = 81.738, $p < 0.01$ | 62.95 (38.30) * | 56.63 (37.52) * | 100.00 (0.00) * |

* Numbers in brackets represent standard deviations of the means.

In all three countries, the farmers cultivate feedstuffs in order to cover the feeding needs of the animals in a more efficient way than just purchasing the necessary feedstuffs from the market. The average cultivated area for feedstuffs is larger in Greece (12.07 ha) and

smaller in Algeria and Tunisia (6.11 and 2.88 ha, respectively). In Tunisia, the cultivated area is totally owned by the farmer, while in Algeria and Greece, the farmers own approximately half of the cultivated area and rent the other half from other individuals (Table 3).

Results from the survey showed that relatively small proportions of the farmers reared other cattle breeds in the past, especially in Algeria (25%) and Greece (19%), indicating a close bond between the farmer and the breed (Chi-square = 224.157 ; df (2); $p < 0.01$; Cramer's V = 0.763). Nevertheless, in Tunisia, the interviewed farmers have chosen to crossbreed the indigenous cattle with commercial cattle breeds in an attempt to improve their productivity.

For the three countries (Algeria, Greece, and Tunisia), the indigenous cattle feeding resources are based on rangelands and pastures with concentrate feed complementation when needed during the critical climatic and physiological periods.

In Algeria, in more than 53% of farms, the cattle flocks are never stabled throughout the year (Figure 2). In Greece, the majority of farmers choose to stable the animals during the night (for protection from predators) and in the winter, while in Tunisia, the animals are stabled only at night. This is probably due to the different climate conditions in the two African countries, where there is no need to confine the animals, since the winter is relatively mild in contrast to Greece (Chi-square = 362.927; df (6); $p < 0.01$; Cramer's V = 0.687).

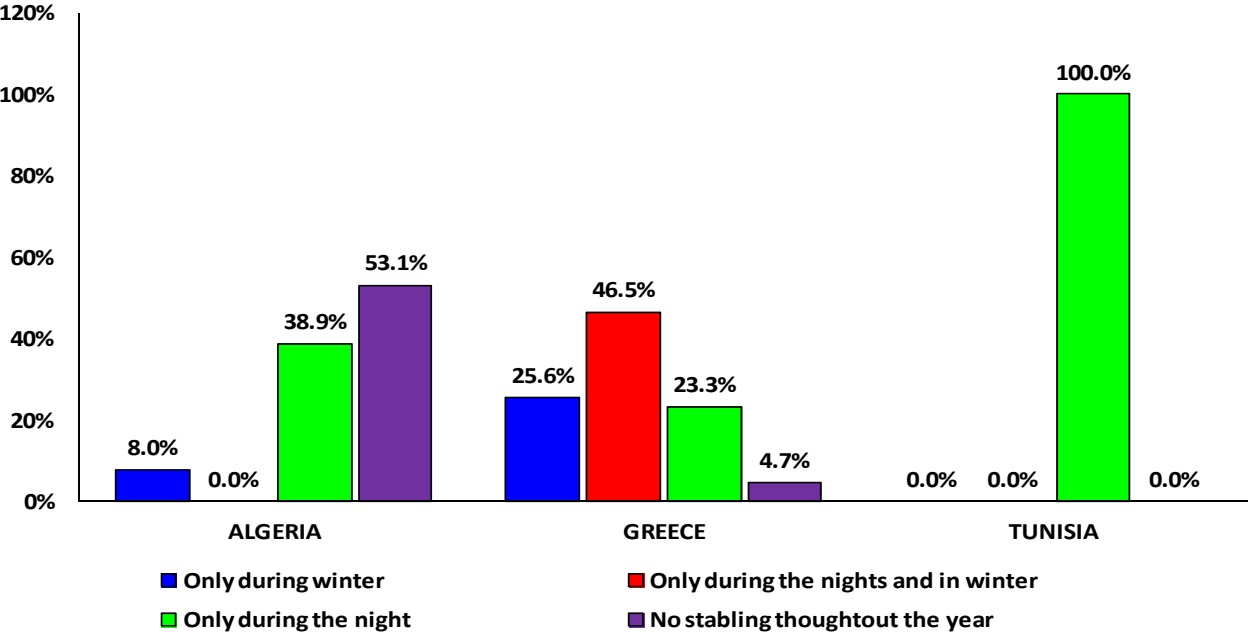

**Figure 2.** Periods of the indigenous cattle stabling in the different study countries (percentage of breeders).

Two types of cattle housings were found in the different study areas: concrete or built housing and loose housing that is more frequent in both Algeria and Greece (Figures 3 and 4). In Tunisia, two types of stables are used, but in 66.5% of the surveyed farms, local and crossbred cattle are housed in concrete stables, and 33.5% of them are housed in loose housing (Chi-square = 35.812; df (2); $p < 0.01$; Cramer's V = 0.306) (Figure 3).

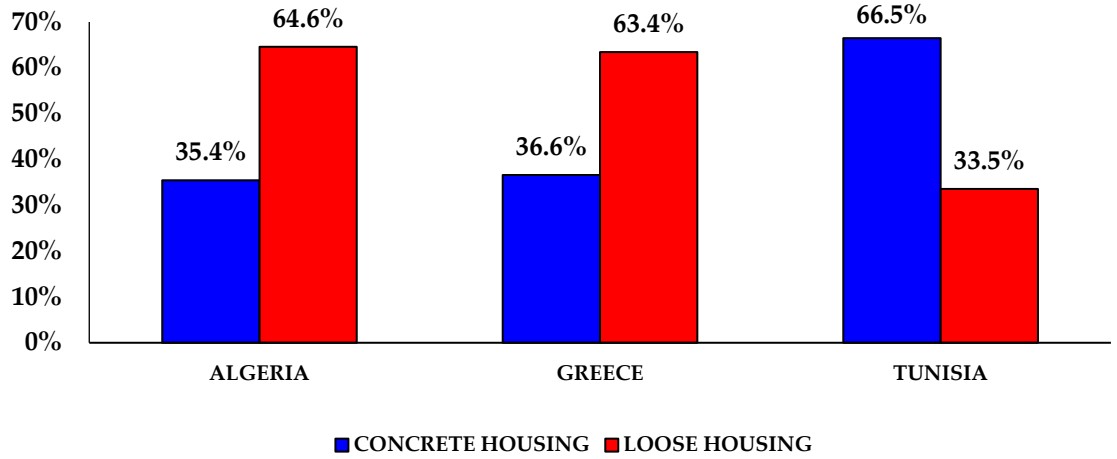

**Figure 3.** Indigenous cattle housing in the study area (percentage of farms).

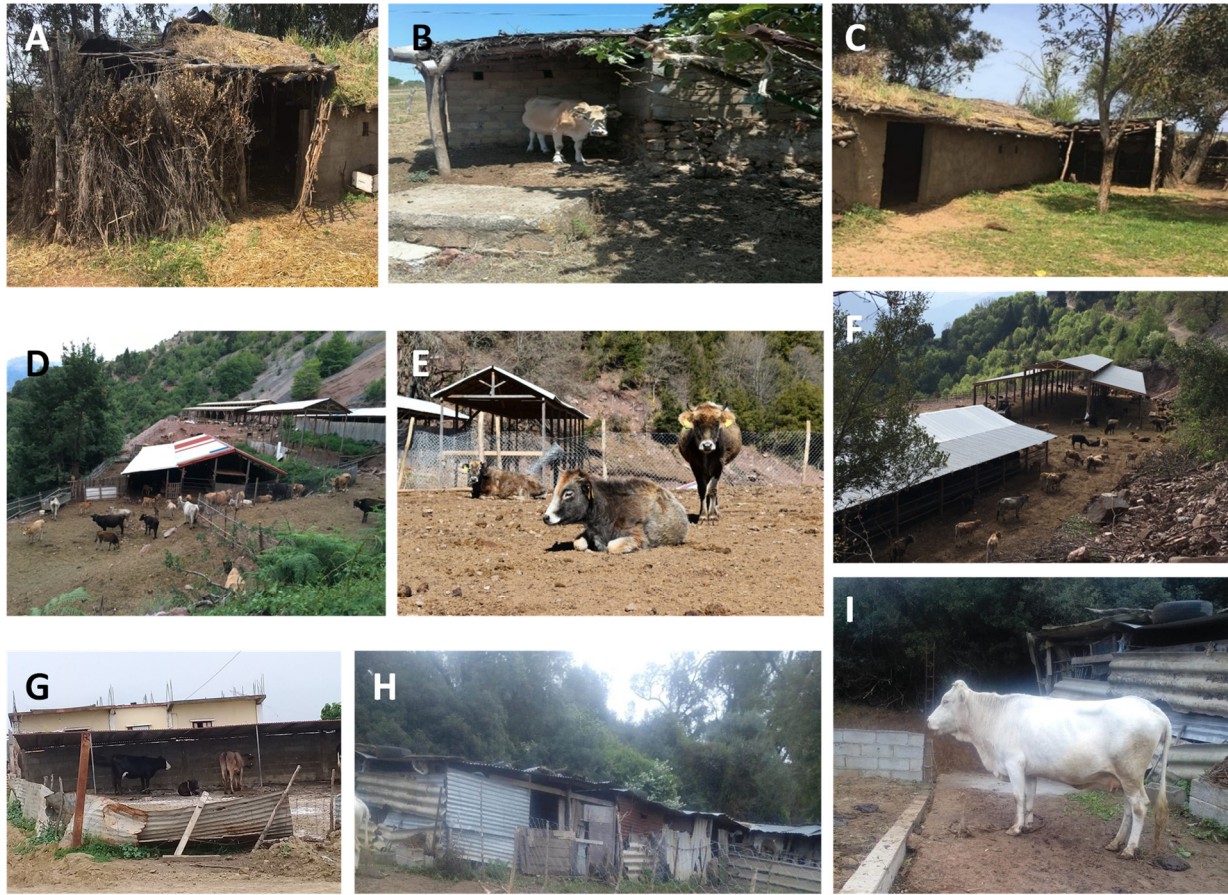

**Figure 4.** Cattle housings: in Tunisia, cattle are housed in concrete stables or in loose housing (**A**–**C**); In Greece, cattle are generally kept in open yard housings (**D**–**F**); In Algeria, cattle housing are generally made of sheet metal and reused wooden planks (**G**–**I**).

### 3.2.2. Labor Force and Farming Practices

In the study area, farming is carried out almost exclusively by the family members with an average size of 1.1, 1.44, and 2.25, respectively, in Algeria, Greece, and Tunisia. External workers are rarely encountered in the interviewed farms. Daily and seasonal tasks differ from one country to another but some of them are practiced in the same way, such as the daily animals' watering and grazing, which is seasonal in 28% of the Greek interviewed farms and daily in Algeria and Tunisia (Chi-square = 32.914; df (2); $p < 0.01$; Cramer's

V = 0.293). Stables' cleaning is generally a daily activity in Tunisia (100%) and seasonal in Greece (81.3%) and in Algeria (87.4%) (Chi-square = 312.196; df (4); $p < 0.01$; Cramer's V = 0.637). Feed supplementation is seasonal in Tunisia (100%) and Algeria (98.3%) and is not applied in the majority of the Greek visited farms (72%) (Chi-square = 388.251; df (6); $p < 0.01$; Cramer's V = 0.710). Milking is also a seasonal task in both Algeria (72%) and Tunisia (100%); nevertheless, it is not practiced in Greece (94%), since the main direction of the production system is meat production (Chi-square = 272.750; df (4); $p < 0.01$; Cramer's V = 0.595). Dehorning and males castrating are never applied on the Tunisian cattle males, which is not the case in Algeria, where these two tasks are almost seasonal, and in Greece, where they are applied only by few farmers (11.6% and 2.3%, respectively, for dehorning and castrating practices).

As showed in Figure 5, animals' allocation in different groups is more practiced by the Algerian surveyed breeders (50%). In Greece, they just separate pregnant cows (14%) or animals that will be fattened (16%). Nevertheless, in Tunisia, no separation is practiced, which is generally related to the farms' small sizes and then to the limited area and feeding resources dedicated to these animals (Chi-square = 338.062; df (22); $p < 0.01$; Cramer's V = 0.765).

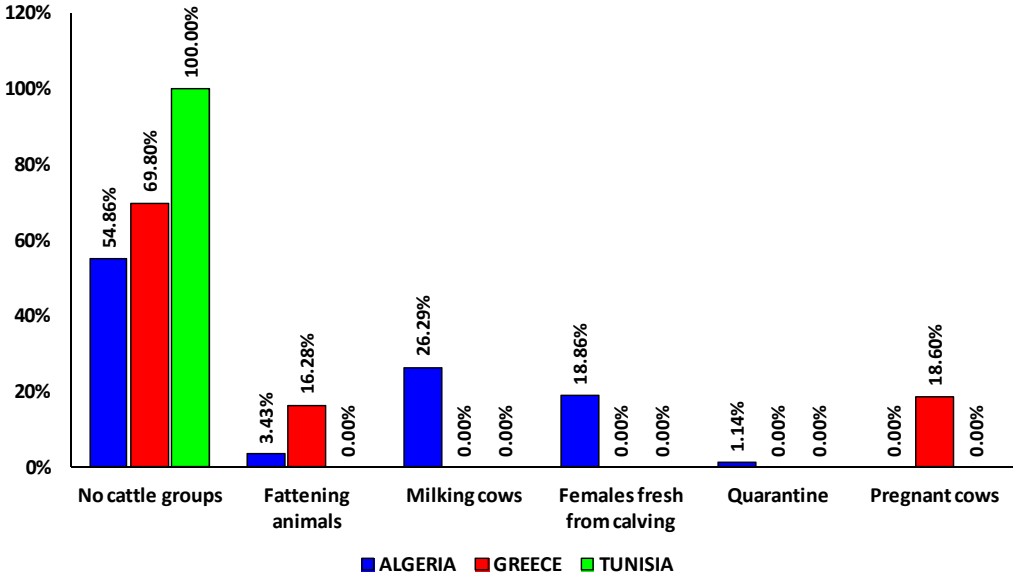

**Figure 5.** Animal's allotment in the surveyed farms.

Only 17% and 14% of the breeders in Algeria and Greece, respectively, declare that they do not purchase feedstuff to meet their animals' needs. In the other farms, feeding management is based on in-farm produced concentrate (32%) and straw (98%) and purchased by-products (44%), concentrate (38%), and straw (44%) in Algeria. Animals' feeding is based on purchased hay (100%), concentrate (100%), and straw (100%) in Tunisia where the main cultivated feedstuff for this kind of production system is the berseem (*Trifolium alexandrinum*) especially in the larger size farms where irrigation is possible. In Greece, the cultivated feedstuff is relatively diversified since alfalfa, corn, triticale, barley, and peas are cultivated respectively by 24%, 33%, 62%, and 28.6% of the interviewed farmers.

After birth, calves take colostrum directly in all the cases in the different study areas (100%). After that, they continue suckling their mothers in all the Greek and Tunisian farms but only in Algeria, they receive colostrum for 24 h after birth and generally continue to receive milk powder (95%) during a period of two months. The average weaning age is about four months and varies from 197.5 days in Greece to 213 days in Tunisia and to 221 days in Algeria (one-way ANOVA $F_{(2, 377)} = 1.572$, $p = 0.209$). Weaning is natural in most cases and forced in respectively 9%, 21%, and 19% of the surveyed farms in Algeria, Greece, and Tunisia (Chi-square = 7.688; df (2); $p < 0.005$; Cramer's V = 0.141).

### 3.2.3. Reproduction and Breeding Management

Reproduction management is basic in the indigenous cattle farms and mainly refers to estrus and pregnancy detection as well as carvings grouping. In Greece, only a few farms use bulls to detect estrus, perform feeding supplementation, and perform calving grouping. In Algeria, almost all the breeders use bulls to detect estrus and also perform pregnancy detection. Nevertheless, in Tunisia, the indigenous and crossbred cattle breeders apply just the estrus detection (Figure 6).

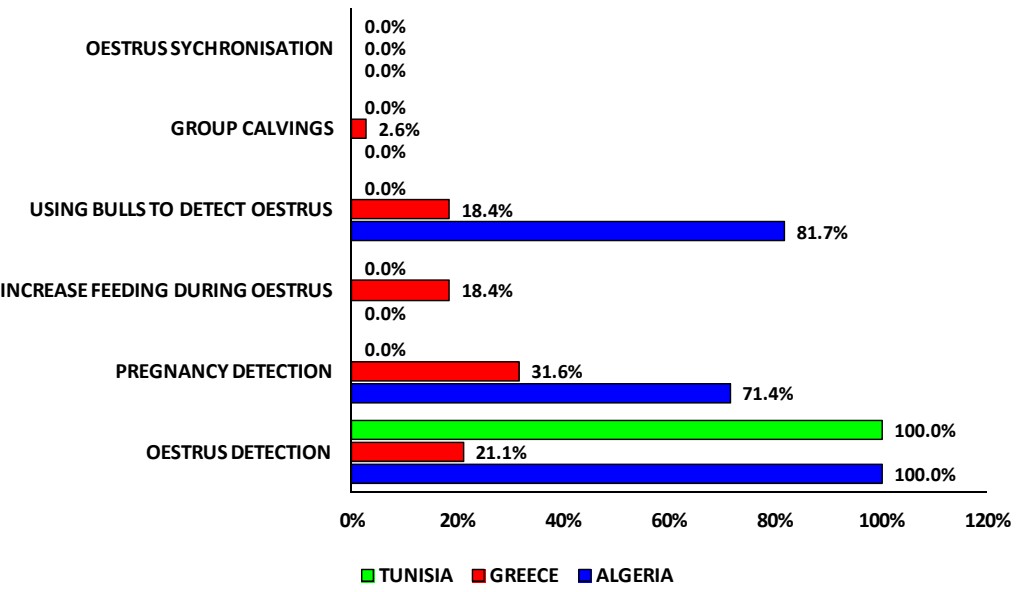

**Figure 6.** Reproduction practices in the surveyed farms.

As shown in Table 4, in the three countries, females enter into reproduction at an average age of 15 months ± 3.5. The average fertility and abortion rates are, respectively, 81.5% and 2%. Calving interval is relatively large (17.5 ± 5 months).

**Table 4.** Reproduction parameters in the surveyed farms.

|  | Age of Heifers' Entry into Breeding (Months) | Fertility Rate (%) | Abortion Rate (%) | Calving Interval (Months) |
|---|---|---|---|---|
| Mean | 14.88 | 81.47 | 1.85 | 17.34 |
|  | (3.41) * | (6.51) * | (3.77) * | (4.98) * |
| N | 375 | 203 | 33 | 381 |

\* Numbers in brackets represent standard deviations of the means.

The majority of the interviewed breeders in Algeria (86%) and Greece (98%) keep more than 18% of the owned cattle females and males for replacement, which was lower in Tunisia (65%), but with no significant differences between the countries (one-way ANOVA $F_{(2, 364)} = 0.920$, $p = 0.399$). Replacement selection criteria are mainly animal phenotype in Tunisia (81%), animal growth performances in Algeria (66%), and parents' performances in Greece (81%), which confirm that the indigenous cattle breeding management and objectives are more or less the same in these three countries (Chi-square = 345.791; df (14); $p < 0.01$; Cramer's V = 0.671).

The basis for every well-designed breeding program is reliable data recording. Unfortunately, in Algeria and Tunisia, data recording in the farm was rarely performed (< 20%), which could be related to the general limited educational level of the indigenous cattle farmers. Nevertheless, in Greece, 78% of the farmers say that they record data on their farms. These data concerned the farm activities (16%), reproduction (84%), health (62.5%), and feedstuff use (91%) in the Greek farms, while in Algeria, only the financial data (100%)

were recorded, and in Tunisia, only information about the farm activities (100%) were recorded. About 62% of the Greek breeders are part of the national performances recording program, and more than 80% of them participate in a genetic resources' conservation program, which is a fact that is rarely seen in the African countries. Nevertheless, in all three countries, the animals' weights are usually estimated visually, and no means of animal identification is used, with the exception of Greece, where all farms use ear tags.

### 3.2.4. Production Systems Directions and Products' Commercialization

In Greece, the local cattle breeds are farmed exclusively for meat production, while in Algeria and Tunisia, the animals are farmed for both meat and milk production in the majority of farms (80.6% and 92.2%, respectively) (Table 5). This could be partly explained by the low meat and milk performances of these breeds, which oblige the farmers to profit from both the milk and meat performances of these flocks in order to increase their revenue.

**Table 5.** Production systems' objectives (percentage of breeders).

| | Objective(s) | Breeding Animals | Meat Production | Milk Production | Meat and Milk Production |
|---|---|---|---|---|---|
| | Algeria | 5.7% | 3.4% | 10.3% | 80.6% |
| Country | Greece | 0.0% | 100.0% | 0.0% | 0.0% |
| | Tunisia | 0.0% | 0.0% | 7.8% | 92.2% |

The lack of a proper selection scheme in smallholder areas results in poor growth rates and possible inbreeding in cattle [23]. In Greece, the average slaughter age of the animals is 18.5 months at an average weight of 309 kg. In Tunisia and Algeria, the respective slaughter age is 17 months and 64 months, respectively, and the average weight is 250 kg and 172 kg, respectively (one-way ANOVA for slaughter age $F_{(2, 370)} = 362.649$, $p < 0.01$; one-way ANOVA for slaughter weight $F_{(2, 337)} = 133.883$, $p < 0.01$).

Periods of animals' commercialization differ from one country to another. In Algeria and Greece, most of the fattened indigenous cattle are commercialized occasionally throughout the year, while in Tunisia, all the interviewed breeders affirm that they sell these animals during the summer, especially for wedding occasions. In Algeria and Greece, 20% of the breeders have sales contracts for animals or products with butchers (94% of the Algerian breeders) or with the animal traders (85% of the Greek breeders) which is not the case in Tunisia, where selling practices are carried out privately between the breeders and their clients (consumers or animal traders) or in the local markets (Chi-square = 560.320; df (10); $p < 0.01$; Cramer's V = 0.916).

Products certification is only encountered in 11.6% of the Greek farms, even though all the interviewed breeders in the three countries believe that their product is of higher quality. In addition, only 17% of the Greek breeders affirmed that the products' prices are affected by their quality. Finally, in Algeria and Tunisia, breeders of the indigenous cattle do not belong to any organizational structure and do not practice any associative activity. Conversely, in Greece, 26.3% of the breeders are part of cooperatives that enable beneficiating from the technical consulting when needed from the cooperative, private, or public entities.

### 3.3. Production Systems Constraints and Improvement Ways

The limiting constraints to the indigenous cattle production systems encountered in Algeria, Greece, and Tunisia are presented in Figure 7, as perceived by the surveyed farmers. All the farmers agreed that a major constraint of the production system is the increased feeding cost (Chi-square = 73.749; df (2); $p < 0.01$; Cramer's V = 0.916). In Algeria and Tunisia, another major constraint is the low productivity of the animals (Chi-square = 298.615; df (2); $p < 0.01$; Cramer's V = 0.884), while both Greek and Tunisian farmers agree that the selling prices are low (Chi-square = 284.436; df (2); $p < 0.01$; Cramer's V = 0.863). The difficult management of the animals is moderately discussed by Greek and Algerian farmers

(Chi-square = 108.588; df (2); $p < 0.01$; Cramer's V = 0.533), while the Tunisian ones care more about the other costs of the production system (Chi-square = 152.661; df (2); $p < 0.01$; Cramer's V = 0.632) as well as for the lack of rules in the market (Chi-square = 251.980; df (2); $p < 0.01$; Cramer's V = 0.812).

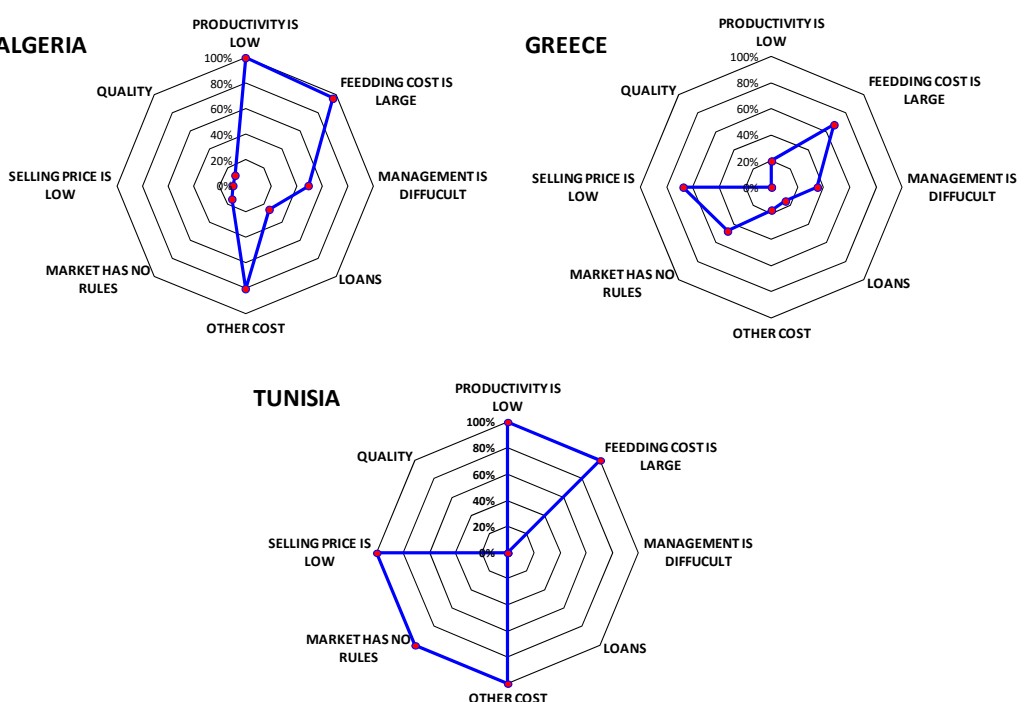

**Figure 7.** Limiting constraints in the studied production systems.

Figure 8 presents the farmers opinions regarding what actions could improve the outcome of the production systems. All farmers agree that higher selling prices (Chi-square = 45.103; df (2); $p < 0.01$; Cramer's V = 0.343) and state funding (Chi-square = 73.896; df (2); $p < 0.01$; Cramer's V = 0.439) would be two major steps in the right direction. In Greece, product certification (Chi-square = 101.108; df (2); $p < 0.01$; Cramer's V = 0.513) and advertisement (Chi-square = 68.196; df (2); $p < 0.01$; Cramer's V = 0.421) are also proposed as solutions while in all the countries, the genetic improvement of animals is moderately appreciated (Chi-square = 100.123; df (2); $p < 0.01$; Cramer's V = 0.511).

### 3.4. Opportunities toward Sustainable Indigenous Cattle Production Systems

Both in Europe and North Africa, indigenous cattle breeds are an essential supplier of food, agricultural power, agrarian culture and heritage, and genetic biodiversity [23]. Indeed, animal genetic diversity allows farmers to set up selection programs in collaboration with the specialized services or to develop new breeds in response to the continuously varying conditions associated to climate change, new or growing disease dangers, new knowledge of human nutritional requirements, and fluctuating market conditions or changing societal needs. It is important to develop concerted, coordinated, and comprehensive farmer training, research, and development programs to address these constraints for the breeders of the indigenous cattle populations that developed their own behavior to adapt with the unstable environmental and economic conditions. An integrated approach with due consideration to proper feeding, breeding, healthcare, and improved management practices are recommended to address the future challenges for sustainable conservation of these native breeds [24]. Subsequently, developing this sector needs both state interventions and adequate farmers' behaviors and activities to be implemented and applied.

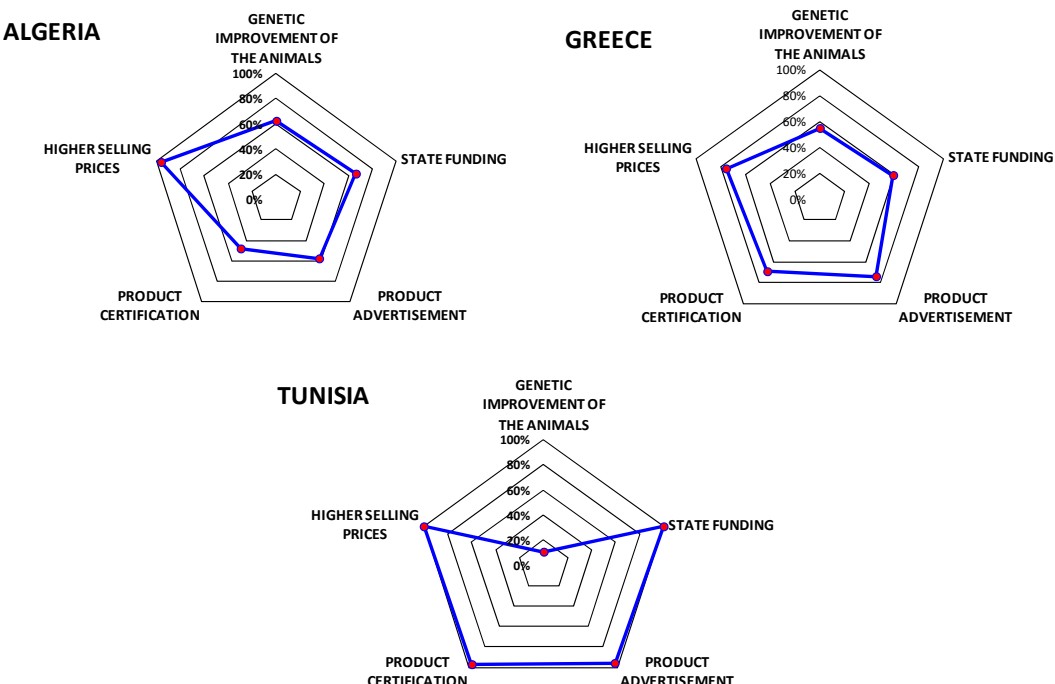

**Figure 8.** Farmers suggested improvement ways of the studied production systems.

### 3.4.1. State Funding

The agricultural policies in general lacked the necessary stakeholder support, both financial and moral, or commitment [25], which is also clear by the small involvement of the state especially in Tunisia and Algeria, where 100% and 66.3%, respectively, of the interviewed farmers stated the need of improvement in the form of state funding. Then, the state must have the leading role in the conservation of animal genetic resources; preserving their legacy for the future generations as a safety belt for the ever-changing environmental conditions is for the greater good. Under this context, it is necessary that the state funds such actions as well as supports financially the farmers that participate in such programs. Nevertheless, the safeguarding of the indigenous cattle breeds through programs and funding is mandatory to include all the relevant actors and foremost the farmers that must actively participate in all the steps. The tools provided should focus not only on conserving the breeds but also improving the management and productivity of the farms as well as preserving the characteristics of the local production systems while implementing current and future innovations. The key action that will ensure the survival of the production systems and the breeds is to find the proper balance between tradition and innovation. The indigenous breeds are highly connected to the local geographical, social, cultural, and economic conditions, and it is in the hands of the state and all the relevant actors to highlight and promote them to future generations.

### 3.4.2. Genetic Improvement of the Animals

The conservation of indigenous cattle breeds is critical for reversing the unprecedented loss of diversity and ensuring the security of cattle genetic resources for economic, eco-logical, and social benefits [26]. Genetic improvement programs must be organized and supervised by the state but implemented by farmers through collective organs. The genetic improvement of animals is a long process with additive results that usually takes years to be quantified and standardized and is important to be treated as such. In the case of indigenous breeds, the main goals would be the following:

- to clearly characterize the breeds and, in cases of crossbred populations,
- to stabilize the breeds,

- to take advantage of their close connection to the local environment and improve their adaptation characteristics, identified, especially in Algeria and Tunisia, as the main reason for choosing the indigenous breeds (72.6% and 98.8%, respectively), and
- to improve their productivity, which was reported by all farmers (100%) as a limiting factor in Algeria and Tunisia, without compromising their unique characteristics. In order to successfully implement genetic improvement programs, the active participation of all the famers of each breed is mandatory in order to have a complete register of the animals, along with accurate recording of their characteristics, performance, and environmental parameters. The responsible state bodies should supervise, coordinate, and control the peripheral actions taken by individual farmers and their collective organs.

An urgent need to establish a conservation plan that includes a well-designed genetic management program for the Tunisian indigenous cattle population was already underlined by [27,28]. This was not expressed by Tunisian breeders (10.8%) that are unaware of the effect of uncontrolled crossbreeding that could easily result in the extinction of the breeds and lack of programs, and when asked about the possible improvement ways, they do not mention the animals' genetic improvement in contrary to farmers in Algeria (62.3%) or in Greece (54.8%). It is important to mention that only farmers in Greece participate in a genetic resources conservation program (80% of the interviewed farmers) and performances recording program (62% of the interviewed farmers).

*3.5. Farmers' Behaviors and Activities*

3.5.1. Farmers Professional Organization

As presented in the results, few farmers in Greece (26.3%) and no farmers in Algeria and Tunisia are part of organizational or professional group. Motivation and sensitization of the breeders to join breeder's cooperatives or associations and farmers training to transform indigenous cattle milk into cheese or other milk derivatives are vital.

3.5.2. Animal Management Practices

As presented in the results, the management practices of indigenous breeds are far from being the optimum for the animals, which is partly due to the traditional character of the production systems. It is indicative that most farmers (96.6% in Algeria, 67.5% in Greece, and 100% in Tunisia) chose feeding costs as a limiting constraint, leaving room for improvement. Additionally, important reproduction practices are not performed in all three countries, with Tunisian farmers only performing estrus detection (100%), a small proportion of Greek farmers (less than 32%) performing increase feeding during estrus and estrus and pregnancy detection, and most of the Algerian farmers performing pregnancy detection (71.4%) and estrus detection (100%). Finally, most of the farmers in all three countries do not allocate animals in different groups (54.8% in Algeria, 69.8% in Greece, and 100% in Tunisia). Thus, small changes in the management practices, without altering the identity of the production systems, could have a great impact on the productivity and the overall welfare of the animals. The changes that are more easily applied, according to the information collected through the questionnaires, are the following:

(a) Improvement of the feeding of the animals with an overall goal to cover the needs of the animal at every production stage with a focus on young animals, females in gestation, and fattening cattle;

(b) Booster feeding before mating could improve the reproduction parameters of the animals;

(c) Control the females' insemination and natural mating;

(d) Better management of newly born calves in terms of feeding and hygiene in order to decrease infant mortality;

(e) Grouping of the animals in critical periods of their life such as gestation and calving.

### 3.5.3. Products Marketing, Certification, and Advertising

It is a common belief among farmers and consumers that the products of indigenous breeds are of high quality, but for most breeds, there is very little evidence that can support such a claim. However, the marketing of these products remains fragmented, as this sector is unsatisfactorily organized. Product certification and product advertisement were suggested as a way of improvement from farmers in all three countries (48.0% and 57.7%, respectively, in Algeria, 69% and 73.8% in Greece, 97% and 95.8% in Tunisia) Thus, it would be for the benefit of all the stakeholders to participate and, if possible, co-fund research projects that would evaluate the quality of the products and promote their unique characteristic and quality. Moreover, as the next step, the farmers could use the results of such projects in order to advertise their products and promote their advantages to the consumers. This would subsequently increase the demand for the products and create a brand name that would be recognizable and desirable by the consumer. As a result, this would also improve the selling prices for the products and the overall income of the farmer. The market promotion of these products will help to incorporate them into a profitable value chain. In this context, more milk and meat quality studies have to be carried out before studying the possibilities of certifying these local products (AOC, IG, AOP, IGP) and labeling, taking into account the consumers' expectations and preferences.

Table 6 presents a SWOT analysis related with the sustainability of the indigenous cattle production systems in the study area, utilizing the data used previously and results and discussion that preceded.

**Table 6.** SWOT analysis of the indigenous bovine production systems toward a sustainable development in Algeria, Greece, and Tunisia.

| | | Strengths | Weaknesses |
|---|---|---|---|
| ✓ | Economic axis | • Indigenous cattle activity is a source of income<br>• Lower dependency on external inputs in marginalized areas | • Absence of financial support to small farmers<br>• Low products' selling prices<br>• Absence of products' certification<br>• Management practices are not optimized |
| ✓ | Social axis | • Breeders attachment to continue practicing this activity<br>• Inherited or/and own investment activity | • Low social value of this activity<br>• Farmers aging<br>• Absence of farmers' training in rearing and farm management techniques |
| ✓ | Environmental axis | • Valuable genetic pool adapted to local and harsh environment<br>• Genetic resources conservation programs | • Problems related to the use of common lands<br>• Over-use of grazing lands<br>• The increase in the inbreeding level<br>• Difficult to monitor animals that graze in far and difficult to access rangelands |
| | | **Opportunities** | **Threats** |
| ✓ | Economic axis | • Distinction of specific products issued from these production systems (milk and meat)<br>• Possibilities of breeders associations' creation and products' prices increase | • Large increase in animals feed<br>• Higher external inputs' prices<br>• Lower profitability of this activity |
| ✓ | Social axis | • Increasing human populations in the marginalized areas<br>• Increasing the employment rate in these zones<br>• Reducing the rural exodus<br>• Consumers' demand for specific zones' products | • High rate of abandonment of this livestock activity<br>• Guidance of farmers to other activities |
| ✓ | Environmental axis | • Considering the eco-systems particularities to advertise the products' quality | • Local cattle breeds are in danger of extinction<br>• Higher level of inbreeding rate |

## 4. Conclusions

The results of the current study show that although smallholder indigenous cattle populations in Algeria, Tunisia, and Greece still exist, their productivity is limited by several constraints that include low performances, limited feed availability, and poor marketing. In addition, the erosion of indigenous cattle populations genetic resources is becoming a serious problem especially in Tunisia. The conservation of these cattle genetic resources could be imperative, as these have been shown to be a useful integral part of agro ecosystems in smallholder areas. The reasons for conserving these flocks vary from their current utilization to the ability to meet future challenges in a dynamic environment. There is a big policy gap in the studied countries, especially Tunisia and Algeria, with regard to the genetic conservation programs. The costs of conservation activities can be met by increasing the market value of indigenous cattle products so that they eventually become self-sustaining. This requires the identification of the beef breeds, their characterization, and the development of marketable products from these breeds. There is a factual necessity to apply breed conservation strategies through securing long-term funding, revamping institutional activities, training technical personnel, and the co-ordination of management efforts, which will promote the conservation of the indigenous cattle populations and improve the sustainable development of these production systems.

**Author Contributions:** Conceptualization and writing—original draft preparation, A.M.-B., D.T., G.K.S.; formal analysis, D.T.; investigation, data curation and validation, A.M.-B., D.T., G.K.S., S.B.S., Y.G., S.S., A.B. (Aissam Boubia), M.A., A.B. (Ali Boudebbouz), and S.B.; review and editing, A.M.-B., D.T., G.K.S., S.B.S., Y.G., S.S., A.B. (Aissam Boubia), M.A. and S.B.; project administration and funding acquisition, G.K.S., S.B.S. and S.B. All authors have read and agreed to the published version of the manuscript.

**Funding:** Breeding and management practices of indigenous bovine breeds: Solutions towards a sustainable future (BOVISOL) is funded through the ARIMNet2 (2017) Joint Call by the following funding agencies: Hellenic Agricultural Organization—Demeter (Greece); Tunisian Ministry of Agriculture, Water Resources and Fisheries represented by the Institution of Agricultural Scientific Research (Tunisia); The Algerian Ministry of Higher Education and Scientific Research and Scientific Research represented by the Directorate General for Scientific Research and Technological Development (Algeria). ARIMNet2 (ERA-NET) has received funding from the European Union's Seventh Framework Program for research, technological development, and demonstration under grant agreement no. 618127.

**Institutional Review Board Statement:** Not applicable.

**Informed Consent Statement:** Not applicable.

**Data Availability Statement:** The data that support the findings of this study are available from the corresponding author, Aziza Mohamed-Brahmi, on request.

**Acknowledgments:** The authors would like to acknowledge the three national funding agencies. Specific acknowledgment is presented to all the breeders and technicians from the different provinces and governorates for their consent and active participation in this study in Algeria, Greece and Tunisia (ODESYPANO).

**Conflicts of Interest:** The authors declare no conflict of interest.

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
