# Peer review of "Challenges and Opportunities of the Mediterranean Indigenous Bovine Populations: Analysis of the Different Production Systems in Algeria, Greece, and Tunisia"

_sustainability, doi:10.3390/su14063356_

Round 1

Reviewer 1 Report

Main Comments: The authors made a large survey in Algeria, Tunisia and Greece to describe farming systems with traditional cattle breeds. The method and concepts are not innovative but this paper provides valuable insight regarding the situation of traditional breeders. I would have appreciated to have more details about rangeland management, animal diets and an idea of the quantity of feedstuff purchased. A better presentation of the production context and trend of pastoralism in these remote areas would also be helpful for the reader. It would also be very interesting to deepen the discussion on  te following challenges

  • Traditional & cultural heritage vs innovation and improved management and performance
  • Improving production level, productivity while conserving robust and low input systems adapted to harsh environment
  • Conservation of the breed, Genetic selection within the breed vs cross breeding

Other Comments:

****Introduction :

* you stated that indigenous breed had « : a valuable locally adapted genetic pool, substantial income to the local economies, and added-value”.. but these systems are declining in favour of more intensive commercial farms : why?

*“ Despite the perfect harmony between these indigenous cattle populations and their natural environment, productivity remains modest “: what is the margin of improvement  for these self-sufficient farms in harsh environment?

****Method:

*How many farms remain at the end? Why have you deleted farms or data with missing values? Since it is not a multivariate analysis, some information, even if partially available could be relevant.

****Results

**3.1 :

“Results show that. Local cattle farming is the main occupation for 82.3% of farmers while 69% of them have a successor in the farm, a number that is greater in the North African countries “strange sentence: why do you opposite main occupation and successor?

*“may be a misinterpretation between performances and robustness of the breeds.” : could you give your definition of performance and robustness. Maybe for farmers robustness and adaptation are indeed criteria of performance!

**3.2 :

table 3/ total area of the farm is lower for Greece than the area cultivated for feedstuff.  How is it possible?  Cattle per ha:the standard deviation is missing. The stocking rate in Greece appears very high. How is it possible? I guess rangelands are not included : how many rangeland do they use?

You say that they “cultivate feedstuffs in order to cover the feeding needs of the animals in a more efficient way than just purchasing the necessary feedstuffs from the market”. Do farmers think that they produce more efficiently feed? Or do you make this assumption? Do you have reference to justify it?

*Why animals are stabled at night during spring and summer in greece:  predators, supplementary feeding, manure storage?

*Feeding practices are not clearly described. I would have appreciated to have ideas of quantities of feed purchased per animal or of proportion of the different type of feed.  

*Figure 7 : estrus instead of oestrus

*You said that “replacement selection criteria was animal phenotype in Tunisia; animal growth performances in Algeria (66%) and parents’ performances in Greece (81%) which confirm that the indigenous cattle breeding management and objectives differ in these three countries.” : for me it is not clear that the objectives of selections are different (phenotype could be growth performance, they may also think that the performance of the next generations depends on grand parents)

*You said that “ in Algeria and Tunisia the animals are farmed for both meat and milk production This could be partly explained by the low meat and milk performances of these breeds which oblige the farmers to profit from both the milk and meat performances of these flocks in order to increase their revenue.” Note that dairy cows always produce meat since they have to produce calves to be milked. But cows with low milk production have often more balanced products between meat and milk.

*I understand that cows are sold on short an local channel: in this case, is labelling so important?

*Figure 8 : production or productivity of the animals?

**3.4

“improve the management and productivity of the farms, as well as preserving the characteristics of the local production systems while implementing current and future innovations” : don’t you think some contradictions exist?

*“unaware of the effect of uncontrolled cross-breeding” : could you explain better?

*“the management practices of local breeds are far from being the optimum for the animals,” “cover the needs of the animal at every production stage”: be aware that optimal is not maximal, given farm constraints and availability of feed, current farming practice could be optimal for them (but not for the society).

**Potential additional references:

Eriksson, C., 2011. What is traditional pastoral farming? The politics of heritage and'real values' in Swedish summer farms (fäbodbruk). Pastoralism: Research, Policy and Practice 1, 1-18

Hadjigeorgiou, I., 2011. Past, present and future of pastoralism in Greece. Pastoralism: Research, Policy and Practice 1, 24

Reviewer 2 Report

This is an important paper on Challenges and opportunities of the Mediterranean indigenous bovine populations: Analysis of the different breeding systems in Algeria, Greece and Tunisia.  The manuscript is reasonably well-written, but

The objective of the summary was proposed to study the local cattle production system and in the introduction, it was proposed to study the breeders of the indigenous cattle populations, management practices, problems and propose solutions to promote sustainability. It is suggested that the authors precisely define the object of study in the research, whether they are the cattle production systems or the native cattle sector, and it is important to specify each terminology (breeding systems, local bovine production systems, local bovine breeds’ farming systems, breeders of the indigenous cattle populations, autochthonous bovine populations).

In the methodology the authors must justify their methodological approach and why only descriptive statistics were used for data analysis, when it was more convenient to use analysis techniques that explain how much variance is due to countries and how much to production units. Multivariate analyses could also have been used.

In the analysis of the problems and opportunities towards sustainable development, the authors are suggested to explain why a theoretical approach was not used and supported by data analysis, so that it does not remain at a descriptive level.

Round 2

Reviewer 2 Report

In the analysis of the problems and opportunities towards sustainable development, the authors were suggested to use data analysis, so that it does not remain at a descriptive level.

Round 3

Reviewer 2 Report

No additional comments

This manuscript is a resubmission of an earlier submission. The following is a list of the peer review reports and author responses from that submission.